# Is fear of falling key to identifying gait and balance abnormalities in community-dwelling older adults? Protocol of a mixed-methods approach

Lewis McColl ,[1] Peter McMeekin,[2] Marie Poole,[1] Steve W Parry[1]

¹Population Health Sciences Institute, Newcastle University, Newcastle upon Tyne, UK
²Faculty of Health and Life Sciences, Northumbria University, Newcastle upon Tyne, UK

**Correspondence to**
Dr Lewis McColl;
l.mccoll2@ncl.ac.uk

## ABSTRACT

**Introduction** The ageing population poses an increasing burden to public health systems, particularly as a result of falls. Falls have been associated with poor gait and balance, as measured by commonly used clinical tests for poor gait and balance. Falls in older adults have the potential to lead to long-term issues with mobility and a fear of falling (FoF). FoF is measured by a variety of instruments; the Falls Efficacy Scale International (FES-I) version is widely used within clinical and research arenas. The ability of the FoF, as measured by the FES-I to predict gait and balance abnormalities (GABAb) has not previously been measured; this study aims to be the first to investigate this prospective relationship.

**Methods and analyses** To investigate the ability of the FES-I to predict GABAb a mixed-methods approach will be used, including quantitative, qualitative and health economics approaches. Initially the ability of the FES-I to identify poor gait and balance will be investigated, along with whether the measure is able to assess change in gait and balance in response to exercise training. The ability of an online FES-I tool to assess poor gait and balance in an alternative pre-existing online strength and balance programme will also be investigated. Interviews will be carried out to investigate participant experiences and motivations of those that are offered Age UK Strength and Balance Training, along with the views of healthcare professionals and Age UK staff involved within the process.

**Ethics and dissemination** NHS REC Approval has been granted (IRAS ID 314705). Study participation is voluntary; participants will be provided with all necessary information within the participant information sheet, with written consent being sought. Study findings will be disseminated through manuscripts in peer-reviewed journals, at scientific conferences and in a short report to participants and the funding body.

## INTRODUCTION

Approximately one-third of community-dwelling adults over the age of 65 fall each year[1] with around half experiencing more than one fall per year.[2]

Falls are commonly associated with gait and balance abnormalities (GABAb),[3] alongside deficits in strength. Falls not only have the potential to cause serious immediate health

## STRENGTHS AND LIMITATIONS OF THIS STUDY

⇒ This study will be the first to prospectively evaluate the ability of the Falls Efficacy Scale International (FES-I) to predict poor gait and balance in community-dwelling older adults.
⇒ This work will also collect participants opinions and motivations of users of Age UK strength and balance training, providing valuable feedback which may be used to improve the service.
⇒ The health economics resource that this work will generate will provide a useful resource for future health economics researchers to use.
⇒ Due to the limited time scale of the study the feasibility of an online FES-I tool to track poor gait and balance of users of the HowFit study will only be completed in a pilot sample.

issues, such as head injuries and/or fractures, but are also a significant source of morbidity and mortality,[4] and may lead to long-term issues with mobility and a fear of falling (FoF).

FoF is a psychosocial construct encompassing multiple falls-related difficulties, including anxiety, loss of confidence and impaired self-efficacy.[5] FoF has also been reported to have a higher prevalence in women, and increases in incidence with age.[6] The measure has also been found to be a predictor of future falls,[7 8] which may be mediated by poor physical performance. The amelioration of FoF through exercise[9 10] further suggests a link between FoF and GABAb, themselves a predictor of future falls, mediating the relationship between FoF and falls in community-dwelling older adults.[11] FoF is measured with variety of instruments, with the Falls Efficacy Scale International (FES-I) version used widely in clinical and research arenas.[12]

Our preliminary study[13] examined routinely collected clinical data of consecutive patients attending the North Tyneside Community Falls Prevention Service (NTCFPS), assessing

the ability of the FES-I to predict GABAb (Gait Speed (GS), Five Times Sit to Stand (FTSS) and Timed Up and Go (TUG)). The NTCFPS is a secondary care service, aiming to provide prevention and management of individuals aged over 60 who are at risk of falling.[14] All patients attending the service are routinely offered strength and balance training through Age UK North Tyneside. Our study found an FES-I score indicative of FoF was associated with scores on commonly used gait and balance measures that suggest both poor physical function and increased falls risks, with good to excellent sensitivity (73%–87.1%) and moderate specificity (50%–69.0%). These are broadly similar to sensitivities and specificities for other commonly used clinical tests including faecal occult blood testing and exercising testing in ischaemic heart disease.[15 16]

Accordingly, our preliminary data show that FoF as measured by the FES-I is associated with scores on commonly used gait and balance tests that indicate a high risk of falling. This association has never been tested in a prospective study. Furthermore, our data highlight the potential of using FES-I as a screening tool to identify community-dwelling older adults at risk of falling who may benefit from strength and balance training rather than relying on physical tests that are rarely performed outside falls clinics and physiotherapy departments. This application may thus have utility both in opportunistic individual screening and community screening programmes.

An alternative to the Age UK strength and balance classes is the HowFit programme, created by the North Tyneside Clinical Commisioning Group. HowFit is an online programme designed to get older adults exercising at home, with three levels of exercises (of increasing difficulty) focused on: Cardiovascular Health; Mobility; Strength; and Stability balance and coordination. The programme currently has over a thousand registered users, and has been advertised on local television and through a mailshot.

The relationship between falls and FoF is complex, with existing evidence showing that exercise (particularly strength and balance training) ameliorates both.[9 17] Identification and treatment of those with GABAb that predispose to falls is critical in falls prevention; utilisation of a tool such as the FES-I has the potential to significantly reduce the time, financial and health burdens of those that fall on local and national healthcare services.

## Research aims
### Primary aim
To investigate the ability of the FES-I to identify GABAb in older adults, along with its validity and sensitivity to assess change in GABAb in response to exercise.

### Secondary aims
► To investigate whether an online FES-I tool appropriate to track FoF as part of the HowFit programme, and can it be used to track improvement.

► To explore the experiences and views of those that are offered Age UK strength and balance training, in particular their motivations, and to explore the views of the Health Care Professionals (HCPs) involved within the process.

► To establish the resources that should be collected to fully assess the impact of fallers on health and social services.

## METHODS AND ANALYSIS
### Study design overview
To investigate the ability of the FES-I to predict GABAb an explanatory mixed-methods study design with a participation selection method will be used, with an initial quantitative focus, followed by a qualitative component.

The study will initially investigate the ability of the FES-I to identify GABAb in older adults, and whether it is a valid and sensitive measure in assessing change in GABAb in response to exercise. Quantitative analysis will also be used to assess the ability of an online FES-I tool as part of the HowFit Programme to measure FoF, and whether it can be used to track improvement. Participant experiences and motivations of those that are offered Age UK strength and balance training will also be collected during qualitative interviews, along with the views of HCPs and Age UK staff involved within the process. Health-related quality of life measurements will also be used to provide a wider picture of participant well-being; these will also be used to aid in completing a conceptual model assessing the resource impact of using the FES-I, creating a resource for future health economics FoF-based research.

### Quantitative methods
#### Strand 1
The first quantitative strand will examine the ability of FES-I to predict GABAb through a prospective cohort study. The principal focus of this strand is investigating the ability of FES-I to predict GABAb, as measured through commonly used gait and balance measures (TUG, GS and FTSS). This strand will involve multiple appointments with participants across a 24-week time period, both in person and remotely via telephone/online. Participants will be community-dwelling adults over the age of 65, who are users of the NTCFPS, in which recruitment will occur. As the project will be carried out through NTCFPS, initial screening and recruitment will be done during standard clinic visits, with participants being consecutive attendees at the NTCFPS. As part of the standard clinic visits, all patients receive a recommendation to attend 10 weeks of Age UK strength and balances classes.

Power calculations carried out were based on sample estimations for a diagnostic study.[18] Sensitivity and specificity estimates were generated based on our previous study,[13] estimating 112 and 703 participants, respectively. Due to the greater importance of sensitivity, an estimated 112 participants were acceptable for the study. Attrition rate in similar studies have been approximately 25%,

meaning an estimated 150 participants are required to achieve a significantly powered sample size.

Prospective participants will then be directed to the researcher (LM), either immediately or arranged within 48 hours via email or phone, after which full informed consent will be taken. FES-I scores and physical function tests relating to GABAb are collected routinely as part of clinic visits, which will be extracted to provide baseline data.

To assess the ability of the FES-I to predict changes in GAB tests participants' adherence (or otherwise) to the Age UK exercise programme, two follow-up appointments will be carried out at 4 and 10 weeks, in which adherence information will be collected. At the conclusion of the Age UK training programme (10 weeks), participants will self-complete the FES-I as part of the evaluation process (carried out as part of Age UK training); participants who do not attend or do not complete the Age UK training will be sent the FES-I via post and asked to return it.

This adherence (or otherwise) will provide a natural grouping of participants; those that attended all classes, those that attended some classes but did not finish the programme and those that declined to attend the classes. Collecting this information will not only allow the ability of the FES-I to predict change in GAB based on strength and balance training to be assessed, but also the ability of FES-I to predict GABAbs pre and post exercise intervention to be investigated. Participants will also be asked to complete a weekly falls diary for the duration of the study, allowing comparison of predicted GABAbs (and subsequently FES-I scores) and incidence and outcomes of falls.

A final research appointment will be carried out at 6 months, in which participants FES-I, physical function and health-related quality of life scores will be recorded, located either at the fall clinic or during a home visit.

During this appointment, participants that attended Age UK sessions will also be asked whether they engaged with any material given at the end of their course (HowFit Booklet and DVD, Balance Exercises and Chair Based Exercise Booklets); all participants will be asked if they sought any information or training themselves.

## Strand 2
The second quantitative strand will be a pilot study in 30 individuals to investigate the feasibility of a future definitive study to assess the use of an online FES-I tool in tandem with the HowFit home exercise programme is predictive of GABAb and improvements in GAB with home directed exercise. This work will not only be used to assess whether the online tool is a feasible method of measuring change in GABAb after strength and balance training, but if the methods used within this study are appropriate in assessing outcomes.

The study will be a prospective observational cohort study, assessing participants over a 6-month period. Participants will once again be community-dwelling older adults, that either self-refer to the HowFit programme,

finding the programme through advertising (search engines, traditional or social media) or routine household leaflet drop. This group will be compared with those that complete the Age UK strength and balance programme from strand 1, providing a natural comparison group. Users of the HowFit programme that give FES-I score of 23 or greater are currently invited to the Falls Clinic for an appointment, and offered Age UK training. This arm will only include those that either do not wish to have an appointment at the falls clinic, or attend the falls clinic and refuse Age UK training and opt to continue using HowFit. If attending the falls clinic participants will have a research appointment as part of their visit, if not an appointment will either be arranged at the falls clinic exclusively for research, or will be carried out at their home. GAB and questionnaires will be completed as in strand 1, with GAB being additionally tested at 4 and 12 weeks, and 6 months.

The online FES-I tool will be developed in collaboration with Tyne Health. The study will follow participants over a 6-month period, with collection of FES-I from the HowFit website. Participants will also be asked to complete a falls diary for the duration of the study, allowing the ability of the FES-I and the HowFit programme to predict the outcome of falls to be tracked.

## Quantitative analysis
Analysis of both quantitative strands will be completed using SPSS 28. Initially descriptive statistics will be generated, which will lead to univariate and multivariate analysis of variance (ANOVA and MANOVA) testing. This will be followed by regression analysis, principally ordinal logistic and negative binomial regression modelling. Appropriate continuous variables will also be manipulated to generate existing, routinely used categorical variables (eg, FES-I score will be translated to those with a low, medium and high FoF), which will be subjected to analysis.

## Qualitative research
The qualitative aspect of the project aims to further the understanding of the implications of using the FES-I to predict GABAb through exploratory interviews. This will focus on the experiences of participants referred on to the Age UK strength and balance programme, particularly in understanding their motivations for reaching the stage they did, and how the programme has impacted on their FoF. Alongside this, the perspective of those who provide the programmes and are involved in the falls clinic process will also be collected.

Up to 30 participants will be interviewed as part of the project (recruited from quantitative strand 1), ideally ensuring that all three previously mentioned groups are represented, increasing the likelihood of data adequacy. Data will be collected through a single qualitative interview with LM, which may be carried out and recorded remotely (Zoom/Teams, over the telephone) or face-to-face dependent on participant preference, along with being in line with current research guidance regarding

COVID-19. These interviews will be carried out 14–16 weeks after the start date of their Age UK strength and balance training, or since the date of recruitment (if not in Age UK training). Healthcare professionals will also be interviewed, who will be relevant professionals associated with the Falls Service and Age UK strength and balance trainers.

Topic guides have been developed by LM and MP to guide the semistructured interviews; initially, a small number of interviews will be carried on both HCPs and patients; these interviews will allow the iterative development of the topic guide, ensuring its effectiveness and reliability. After these interviews, the remaining participants and HCPs will be interviewed, and if required the topic guides will be updated. University approved transcription services will be used.

## Qualitative analysis
Analysis of interviews will be guided by theoretical frameworks, drawing on existing theories and methodologies to evaluate complex interventions, principally thematic analysis and realist review methodology.[19] Analysis will be completed by using NVivo qualitative data analysis software, with coding being completed by LM.

## Participants
Participants will be included within any aspect of the quantitative or qualitative study if they are (A) at least 60 years of age, (B) are not receiving any care assistance and (C) have an FES-I of above 16.

Participants will be ineligible if they (A) have any neurological, musculoskeletal or cardiovascular diseases that prevent them from completing GABT, (B) have a progressive neurological condition (eg, Parkinson's disease, multiple sclerosis), (C) have a recent history (<6 months) of orthopaedic fracture or joint replacement and (D) a history of osteoporosis-related fractures (low impact hip, ankle or wrist fractures).

To be involved in quantitative strand 2, participants who have been contacted by the falls clinic will either have to reject the clinic appointment, or attend the clinic appointment and reject Age UK training. Participants who are invited to attend the falls clinic (after scoring higher than 23 on the FES-I) and accept Age UK training will not be accepted for the study.

Participants who do not speak English will only be excluded from the study if a translator cannot be found (if a prospective participant is hearing impaired and can sign, a sign-language interpreter must be found).

## Health economics
The project will also have a complimentary health economics aspect. EuroQol 5 Dimension (EQ-5D) and Icepop capability measure for older people (ICE-CAP-O) data will be collected prospectively from quantitative participants as part of their baseline and final clinic visits to provide valid and reliable measures of participants health-related quality of life and perception of their well-being. This data aid in the development of conceptual and logic models, detailing links between FES-I and falling, addressing a gap in the literature and providing a resource for future health economics projects in the subject area.

As EQ-5D and ICECAP O data will be collected at baseline and 24 weeks the ability of Age UK training, HowFit training, or otherwise to improve health-related quality of life will be investigated, along with correlations between this relationship and FES-I scores.

## Patient and public involvement
Two patient and public involvement groups were involved in the study design process. The groups were made up of current participants of the Age UK strength and balance classes, and a group from the North Tyneside Patients Forum. Feedback was sought initially on overall study design, with further meetings reviewing participant information sheets, qualitative topic guides and further study documents.

## Ethics and dissemination
Study participation is voluntary with participants receiving no compensation, but being reimbursed for travel costs to final research appointments. Appropriate research ethics has been approved (NHS REC- IRAS ID:314705). Participants will be provided with all necessary information within the participant information sheet, with written consent being sought after. The study presents a low risk to participant health. Whether or not a prospective participant decides to take part in the research will not affect the care receive through the falls clinic or Age UK training. Study findings will be disseminated through manuscripts in peer-reviewed journals, at scientific conferences and in a short report to participants and the funding body. Data will be managed securely, and only distributed to researchers after a successful information request.

**Acknowledgements** This study was funded by the NIHR NENC ARC.

**Contributors** LM, PM, MP and SWP contributed to study inception and design and manuscript reviewing. LM drafted the manuscript.

**Funding** This study is funded by the National Institute for Health Research (NIHR) Applied Research Collaboration (ARC) North East and North Cumbria (NIHR200173).

**Disclaimer** The views expressed are those of the author(s) and not necessarily those of the NIHR or the Department of Health and Social Care.

**Competing interests** None declared.

**Patient and public involvement** Patients and/or the public were involved in the design, or conduct, or reporting, or dissemination plans of this research. Refer to the Methods section for further details.

**Patient consent for publication** Not applicable.

**Provenance and peer review** Not commissioned; externally peer reviewed.

**ORCID iD**
Lewis McColl http://orcid.org/0000-0001-7448-0113

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
