## [Reviewer comments · BMJ Open]

ARTICLE DETAILS

TITLE (PROVISIONAL)	Is Fear of Falling key to identifying gait and balance abnormalities in community dwelling older adults? Protocol of a mixed methods approach.
AUTHORS	McCull , Lewis; Mcmeekin, Peter; Poole, Marie; Parry, Steve W

VERSION 1 – REVIEW

REVIEWER	Negarandeh, Reza Univ Tehran Med Sci
REVIEW RETURNED	13-Aug-2022

GENERAL COMMENTS	Thank you for submitting this good paper to the BMJ Open. The manuscript is about the protocol of mixed methods study that its Primary Aim is to investigate the ability of the FES-I to identify GABAb in older adults, along with its validity and sensitivity to assess change in GABAb in response to exercise. while it is a well-designed study that investigates a good question on this important topic, I have some questions and comments that need to be addressed before publication. Please express the aim of the study clearly in the abstract. On page 6 line 50 you wrote that "The quantitative aspects of the PhD will be complimented by a qualitative component, aiding...." the sentence is not clear. please clarify it. for exploring the perspectives of different groups of participants It is necessary to recruit enough number of them in the qualitative phase. I expect that the interview guide was included in the protocol. I can not understand the link between the Health Economics phase and the two previous phases. as well in the mixed methods studied we use a method for incorporating the results of different phases. But I cannot find anything about this important issue in the manuscript.
---

REVIEWER	Martins, Anabela Institute Polytechnic of Coimbra, Escola Superior de Tecnologia da Saúde de Coimbra - Physiotherapy
REVIEW RETURNED	15-Aug-2022

GENERAL COMMENTS	Thank you very much for the opportunity to review the manuscript. The manuscript is relevant. However, I suggest revision of some aspects before acceptance. All notes or comments that authors can improve are highlighted in the manuscript. - The reviewer provided a marked copy with additional comments. Please contact the publisher for full details.
---

REVIEWER	Manlapaz, Donald University of Santo Tomas, Department of Physical Therapy- College of Rehabilitation SSciences
REVIEW RETURNED	27-Sep-2022

GENERAL COMMENTS	Overall, the study protocol is feasible but needs some clarity. 1. Please indicate on your methods that you will utilise an explanatory sequential mixed-methods study with a quantitative followed by a qualitative strand. 2. How will you conduct the integration of the analysis of data? Is it quanti first then quali or at the same time? 3. Are both strands with equal weight (quali=quanti)?
---

VERSION 1 – AUTHOR RESPONSE

Reviewer: 1

Dr. Reza Negarandeh, Univ Tehran Med Sci

Comments to the Author:

Please express the aim of the study clearly in the abstract. **“this study aims to be the first to investigate this prospective relationship.”**

On page 6 line 50 you wrote that "The quantitative aspects of the PhD will be complimented by a qualitative component, aiding...." the sentence is not clear. please clarify it. **Accepted and Changed-clarifications/alterations made, ensuring criteria is met.**

for exploring the perspectives of different groups of participants It is necessary to recruit enough number of them in the qualitative phase. **This is something we have a contingency for if not enough are recruited to Quant1 who wish to do further research; we plan on recruiting immediately from the Falls Clinic for the interview, however this is unlikely to occur.**

I can not understand the link between the Health Economics phase and the two previous phases. as well in the mixed methods studied we use a method for incorporating the results of different phases. But I cannot find anything about this important issue in the manuscript. **Accepted and Addressed- whilst part of my PhD, the majority of the health economics aspects are not directly relevant to the study so have been removed.**

Reviewer: 2

Dr. Anabela Martins, Institute Polytechnic of Coimbra

Comments to the Author:

Thank you very much for the opportunity to review the manuscript.

The manuscript is relevant. However, I suggest revision of some aspects before acceptance. All notes or comments that authors can improve are highlighted in the manuscript attached. **Comments accepted and changed.**

Reviewer: 3

Dr. Donald Manlapaz, University of Santo Tomas

Comments to the Author:

Overall, the study protocol is feasible but needs some clarity.

1. Please indicate on your methods that you will utilise an explanatory sequential mixed-methods study with a quantitative followed by a qualitative strand. **Accepted and changed.**
2. How will you conduct the integration of the analysis of data? Is it quanti first then quali or at the same time? **Qual participants are those that have been recruited in Quant1, so for each participant it will be quant-qual. There may be an overlap between some qual finish and quant start.**
3. Are both strands with equal weight (quali=quanti)? **Although the body of work itself is mixed methods, the strands will be considered individually- data from Quant will not be used with responses in Qual.**

VERSION 2 – REVIEW

REVIEWER	Negarandeh, Reza Univ Tehran Med Sci
REVIEW RETURNED	05-Nov-2022
GENERAL COMMENTS	Thank you for considering my comments; in my opinion, the revisions improved the manuscript substantially, and it is suitable to publish now.
REVIEWER	Martins, Anabela Institute Polytechnic of Coimbra, Escola Superior de Tecnologia da Saúde de Coimbra - Physiotherapy
REVIEW RETURNED	01-Nov-2022
GENERAL COMMENTS	The reviewer provided a marked copy with additional comments. Please contact the publisher for full details.
REVIEWER	Manlapaz, Donald University of Santo Tomas, Department of Physical Therapy- College of Rehabilitation Sciences
REVIEW RETURNED	31-Oct-2022
GENERAL COMMENTS	All comments and feedback on my part were addressed appropriately.